# *PLAG1* g.8795C>T Mutation Regulates Early Body Weight in Hu Sheep by Weakening miR-139 Binding

**DOI:** 10.3390/genes14020467

**Published:** 2023-02-11

**Authors:** Yue Wang, Yin-xia Li, Jun Zhang, Yong Qian, Chun-hua Meng, Ji-feng Zhong, Shao-xian Cao

**Affiliations:** 1Institute of Animal Science, Jiangsu Academy of Agricultural Sciences, Nanjing 210014, China; 2Jiangsu Provincial Engineering Research Center of Precision Animal Breeding, Nanjing 210014, China; 3Key Laboratory of Crop and Animal Intergrated Farming, Ministry of Agriculture and Rural Affairs, Nanjing 210014, China

**Keywords:** birth weight, Hu sheep, miR-139, *PLAG1*, SNPs, weaning weight

## Abstract

Sheep birth and weaning weights indicate their growth and survival. Thus, identifying molecular genetic markers for early body weight is important in sheep breeding. Pleomorphic adenoma gene 1 (*PLAG1*) is important for regulating birth weight and body length in mammals; however, its relationship with sheep body weight remains unknown. Here, the 3′-untranslated region (3′-UTR) of the Hu sheep *PLAG1* gene was cloned, single nucleotide polymorphisms (SNPs) were screened, genotype–early body weight relationships were analyzed, and the possible molecular mechanism was explored. *PLAG1* 3′-UTR sequences with five forms of base sequences plus poly(A) tails were detected in Hu sheep and the g.8795C>T mutation was identified. Luciferase reporter assay indicated that the g.8795C>T mutation influenced *PLAG1* post-transcriptional activity. miRBase prediction showed that the g.8795C>T mutation was located in the miR-139 seed sequence binding region, and miR-139 overexpression significantly decreased both *PLAG1*-CC and *PLAG1*-TT activities. Moreover, the luciferase activity of *PLAG1-*CC was significantly lower than that of the *PLAG1-*TT, but miR-139 inhibition substantially increased both *PLAG1*-CC and *PLAG1*-TT luciferase activities, suggesting that *PLAG1* is the target gene of miR-139. Thus, the g.8795C>T mutation upregulates *PLAG1* expression by weakening its binding with miR-139, promoting *PLAG1* expression, and increasing Hu sheep birth and weaning weights.

## 1. Introduction

Growth traits are important economic traits of sheep [1,2]. Birth weight and weaning weight, which are closely related to the weight of sheep at each stage and the lamb survival rate under the same nutrition and environmental conditions [3], are the key breeding indicators of mutton sheep. Body weight is a crucial selection criterion for improving livestock production. Animal weight significantly affects farmer or breeder income [4,5].

Pleomorphic adenoma gene 1 (*PLAG1*), a proto-oncogene encoding a zinc finger containing a transcription factor [6], is a “fetal” expression gene, which is highly expressed in fetal tissues but lowly expressed in adult animals [7,8]. Zheng et al. [9] found that the expression of the *PLAG1* gene in the liver of adult rats was lower than that in rats at embryonic stage 14. Moreover, the *PLAG1* gene has not been detected in the brain, thymus, stomach, intestine, spleen, prostate, kidney, uterus, tongue, lung, liver, and other organs of adult mice but has been detected at high levels in embryonic mice, and its postnatal expression level is low [10]. Another study found that *PLAG1* expression levels were significantly different between 5-month-old fetuses and 36-month-old adult cattle, and that *PLAG1* was primarily expressed in the embryonic tissues of Chinese cattle [11].

As a transcriptional factor, PLAG1 plays an important role in regulating body weight and body length in mice, cattle, pigs, and humans [12]. Compared with wild-type mice, *PLAG1* knockout mice weigh less at birth, 11.5 days into the embryonic stage, with a weight difference of up to 18%. By the end of pregnancy, the body weight of *PLAG1* knockout mice is 30% lower [10]. In cattle, Karim et al. [13] found that the A/G mutation at the 12 bp position upstream of a variable number tandem repeat in the promoter region of the *PLAG1* gene is linked to the (CCG)n copy, which is involved in regulating body weight and height. Furthermore, Littlejohn et al. [14] reported that variations in the *PLAG1* gene are related to early body weight and can affect the weight of puberty-stage cows. Mutations identified in the *PLAG1* gene are also associated with mammalian growth traits. For example, Abi et al. found that a genetic defect in the HMGA2–PLAG1–IGF2 pathway could restrict the growth of the fetus and mammals post-birth and that the *PLAG1* gene plays an important role in this pathway [7]. Additionally, *PLAG1* single nucleotide polymorphisms (SNPs) in the chromosomal region were found to be related to human height in a Korean study with a sample size of 8,842 [15], which was verified by Okada et al. [16], who found that PLAG1 was associated with height in Japanese adults. In cattle, SNPs in the *PLAG1* gene influence birth weight. Fink et al. [17] found that the G>T mutation at the rs109815800 locus of *PLAG1* causes differential expression and is related to cattle weight, and the G allele increases the body height of local Chinese cattle. In pigs, PLAG1 was proximal to the top SNP and stood out as a strong candidate gene associated with growth and fatness in White Duroc × Erhualian F2 population and a Chinese Sutai half-sib population [18]. PLAG1 was identified as a candidate gene for body weight and backfat thickness in 365 individuals of the Chinese Sujiang pigs [19].

However, little is known about the genetic characteristics and polymorphisms of the *PLAG1* gene in sheep. Therefore, this study identified the complete 3′-untranslated region (3′-UTR) of the ovine *PLAG1* gene and screened for relationships between polymorphisms and early body weight using DNA sequencing. Furthermore, miRNA prediction and corresponding luciferase reporter assays were conducted to confirm and explain changes in molecular mechanisms because the SNPs of the *PLAG1* gene are involved in regulating early body weight in Hu sheep. Combining these results, we speculate that the SNP locus (g.8795C>T mutation) weakens the binding capacity between miRNA (miR-139) and *PLAG1*, further increasing the post-transcriptional activity of *PLAG1* and upregulating the early weight of Hu lambs.

## 2. Materials and Methods

### 2.1. Samples

Hu sheep were obtained from Xilaiyuan Sheep Industry Co., Ltd., Jiangsu Province, China. After the sheep were slaughtered, muscles were obtained, immediately placed in liquid nitrogen, and then stored at −80 °C until the total RNA was extracted and reverse-transcribed using an RNA extraction kit (Biotaka, Nanjing, China) and reverse transcription kit (Vazyme, Nanjing, China), respectively, for gene cloning and rapid amplification of cDNA ends (RACE) (Takara, Nanjing, China). The ear tissues of 352 lambs of the same herd, having the same environment and diet, and similar ages at the same season, were collected from the Xilaiyuan Sheep Industry Co., Ltd., along with corresponding records of birth weight and weaning weight (45 d after birth). DNA was extracted using a standard phenol–chloroform extraction protocol. All experiments were performed following the protocol approved by the Committee on Ethics of Animal Experimentation of the Jiangsu Academy of Agricultural Sciences (No. 63 of the Jiangsu Academy of Agricultural Sciences, approved on 8 July 2014).

### 2.2. Primer Designing

Primers (Table 1) based on ovine PLAG1(XM_042254231.1), β-Actin (NM_001009784), and miR-139 (NC_040266.1) sequences from the NCBI database were designed using Primer Premier v5.0 (PREMIER Biosoft, California, USA). All primers used for RACE, PCR amplification, real-time PCR, and plasmid construction were synthesized by Tsingke Biological Technology (Tsingke, Beijing, China).

### 2.3. Rapid-Amplification of cDNA Ends (RACE)

A SMARTer RACE 3′ kit (Takara, Nanjing, China) was used to amplify the end sequence of *PLAG1* 3′-UTR. The 3′ RACE method combined with nested polymerase chain reaction (PCR) was used to obtain the full-length *PLAG1* 3′-UTR. The 3′ RACE protocol was as follows: the first round of the nested PCR protocol was at 95 °C for 3 min, followed by 30 cycles at 95 °C for 15 s, 68 °C for 15 s, and 72 °C for 3 min, and a final round at 72 °C for 5 min; the second round of the nested PCR protocol was at 95 °C for 3 min, followed by 30 cycles at 95 °C for 15 s, 56 °C for 15 s, and 72 °C for 3 min, and a final round at 72 °C for 5 min. Amplification was visualized using 1.5% agarose gel electrophoresis and ligated into pMD19-T vector and transformation, and then the bacterial liquid was sequenced (Tsingke, Beijing, China).

### 2.4. Cloning, Sequencing, and Genotyping

Seven primers (P1-P7) were designed to amplify the 3′-UTR sequence of the *PLAG1* gene using PCR (Table 1), which was performed in a total volume of 50 µL containing 2 µL of RT production or 100 ng DNA, 25 µL 2 × Taq Master Mix (Vazyme, Nanjing, China), and 2 µL each of 10 nmol/L upstream and downstream primers (Table 1). The amplification conditions were as follows: 95 °C for 3 min; 35 cycles at 95 °C for 15 s, 54–57 °C for 15 s, and 72 °C for 40 s; final extension at 72 °C for 5 min.

We screened the *PLAG1* 3′-UTR sequences for SNPs using a DNA pooling sequencing assay with primer P6 (Table 1). A total of 5 µL of 100 ng/µL DNA was collected from five lambs and pooled. The PCR products were sequenced in both directions. SNPs were identified using Chromas v2.31. The SNPs were genotyped by sequencing.

### 2.5. Real-Time PCR

A stem-loop primer (P18 in Table 1) was used for specific cDNA synthesis of miR-139. The relative expression levels were normalized to those of U6 (P13 in Table 1) or β-actin (P11 in Table 1). Real-time PCR was performed using ChamQ SYBR qPCR master mix (Vazyme, Nanjing, China) in a reaction volume of 20 µL and an ABI Step One system (Applied Biosystems, Carlsbad, USA). For the genes, the 20 μL reaction volume contained 1 μL of cDNA, 0.4 μL each of forward and reverse primers (10 μM), 10 μL of 2 × ChamQ universal SYBR qPCR Master mix (Vazyme, Nanjing, China), and 8.2 μL of nuclease-free H_2_O. Thermocycling conditions were as follows: 95 °C for 30 s, 40 cycles at 95 °C for 5 s, and 60 °C for 30 s, followed by 95 °C for 15 s, 60 °C for 60 s, and 95 °C for 15 s. For miR-139, the 20 μL reaction volume contained 1 μL of cDNA, 0.4 μL of specific primer (10 μM), 0.4 μL mQ primer R (10 μM), 10 μL of 2 × miRNA universal SYBR qPCR Master mix (Vazyme, Nanjing, China), and 8.2 μL of nuclease-free H_2_O. Thermocycling conditions were as follows: 95 °C for 30 s, 40 cycles of 95 °C for 5 s, and 60 °C for 30 s, followed by 95 °C for 15 s, 60 °C for 60 s, and 95 °C for 15 s. Each sample had at least three replicates to ensure accuracy. Relative expression was calculated using the 2^−ΔΔct^ method. Table 1 lists all the real-time PCR primers.

### 2.6. Prediction of miRNA Binding Site

MiRBase (www.mirbase.org, accessed on 7 March 2019) online software was used to predict the miRNA-binding site of the *PLAG1* 3′-UTR sequence. The steps were as follows: paste the *PLAG1* 3′-UTR sequence to search for similarity with the miRbase miRNA sequence and choose to search against mature miRNAs; the E-value cutoff was 10 and the maximum number of hits was 100.

### 2.7. Plasmid Construction

In order to identify whether the g.8795C>T mutation affects the luciferase activity of *PLAG1*, a fragment containing the g.8795C>T mutation was amplified and cloned into the XhoI and SalI (Takara, Nanjing, China) sites of a pmirGLO-basic vector to create pmirGLO-TT and pmirGLO-CC vector, respectively. Mimics and inhibitors of miR-139 and corresponding NC were then synthesized by GenePharma (Shanghai, China). Plasmids were extracted using an Endo-free Plasmid Mini Kit (OMEGA, Cambridge, USA).

### 2.8. Transfection and Dual-Luciferase Reporter Assay

First, 293T cells were seeded into 24-well plates and maintained at 37 °C and 5% CO_2_ in Dulbecco’s modified Eagle’s medium (DMEM; Gibco, California, USA) with 10% fetal bovine serum. Then, luciferase reporter plasmids of *PLAG1* 3′-UTR were transfected into the 293T cell lines using Lipofectamine 3000 (Invitrogen, California, USA) according to the manufacturer’s instructions. The pRL-TK *Renilla* luciferase reporter plasmid was used as an internal control. Then, 48 h after transfection, the cells were lysed, and the lysates were harvested and subjected to a luciferase assay using a dual-luciferase reporter assay system (Promega, Madison, USA) to measure the firefly and *Renilla* luciferase activities. Luciferase activity was normalized to the firefly/Renilla ratio. All remaining steps were performed as described previously [20].

### 2.9. Statistical Analyses

The correlation analysis model was identified using the general linear model (GLM) procedure in SPSS software (version 16.0; SPSS Inc., Chicago, USA). The influencing factors in this study included birth status (simple or twin), maternal age (two levels: two or three years old), sex (male or female), and genotype (CC or CT); Yijkl = μ + Gi + Bj + Mk + Sl + eijkl, where Yijkl is the trait measured for each ijk animal, μ is the overall population mean, Gi is the fixed effect associated with genotype, Bj is the fixed effect of birth status, Mk is the fixed effect of maternal age, Sl is the fixed effect of sex, and eijkl is the random residual error.

Allele frequencies were calculated using PopGene v1.31. All results are reported as mean ± standard error of the mean (SEM). Statistical analysis was performed with independent sample *t*-tests using SPSS software (version 16.0; SPSS Inc., USA). Multiple comparisons of the different groups were conducted using a one-way analysis of variance. Statistical significance was set at *p* < 0.05.

## 3. Results

### 3.1. 3′-UTR Identification and Characteristics of Ovine PLAG1 Gene

To identify the complete 3′-UTR sequence of the ovine *PLAG1* gene, the partial sequence following the stop codon of the *PLAG1* gene was cloned, and the length of the 6670 bp sequence was obtained by segmented amplification using seven pairs of primers (Figure 1a). Based on the obtained sequence of *PLAG1*, P8, and P9, primers shown in Table 1 were designed to amplify the 3′-UTR end sequence using 3′ RACE. Two distinct bands, named A and B, were amplified using nested PCR (Figure 1b). Sequencing analysis indicated that only the B band was the sequence of the *PLAG1* 3′-UTR, which included five spliceosomes with different base sequences plus poly A tails: TCTAC, TCTACC, TCTACTGAAGATGTTTC, TCTACTGAAGATGTTTT, and TCTACTGAAGATGTTTTGC (Figure 2).

### 3.2. SNP Screening in the 3′-UTR of the Ovine PLAG1 Gene

Primers were designed to screen for SNPs by DNA pool sequencing, and g.8795C>T was identified in the 3′-UTR region of the *PLAG1* gene. In the Hu lamb population, three genotypes (CC, CT, and TT) were identified at the g.8795 locus (Figure 3a) with genotype frequencies of 0.8466 (298/352), 0.1477 (52/352), and 0.0057 (2/352), respectively (Figure 3b). The lambs with the CT genotype exhibited higher birth weight (3.42 ± 0.08) and weaning weight (13.51 ± 0.39) than those with the CC genotype (3.13 ± 0.03, *p* < 0.05 and 12.67 ± 0.17, *p* > 0.05, respectively) in a population of 352 Hu sheep. Further analysis revealed that the birth and weaning weights of different genotypes at the g.8795 locus were also significantly different between ram lambs and ewe lambs. Specifically, ram lambs with CT genotypes exhibited higher birth weight and weaning weight (3.56 ± 0.11 and 15.54 ± 0.48, respectively) than those with CC genotypes (3.22 ± 0.05 and 14.39 ± 0.20, respectively, *p* < 0.05) among 168 ram lambs, and ewe lambs with the CT genotype exhibited higher birth weight (3.29 ± 0.10) and weaning weight (12.14 ± 0.42) than those with the CC genotype (3.02 ± 0.04 and 11.19 ± 0.19, respectively; *p* < 0.05) among 191 ewe lambs. Furthermore, twin lambs with the CT genotype exhibited higher birth weight (3.53 ± 0.11) and weaning weight (13.32 ± 0.55) than those with the CC genotype (3.26 ± 0.05, *p* < 0.05 and 13.14 ± 0.23, *p* > 0.05, respectively) among 177 twin lambs.

### 3.3. Effect of g.8795 C>T Mutation on the Post-Transcriptional Activity of the Ovine PLAG1 Gene

To investigate whether the g.8795 C>T mutation in the 3′-UTR of the *PLAG1* gene affected post-transcriptional activity, the g.8795 C and g.8795 T-type 3′-UTR regions were amplified and cloned into a pmirGLO vector, resulting in pmirGLO-CC and pmirGLO-TT, respectively, which were transfected into 293T cells. The results indicated that the luciferase activity of plasmids pmirGLO-TT and pmirGLO-CC was significantly lower than that of the control group (*p* < 0.01), and the luciferase activity of plasmid pmirGLO-TT was higher than that of plasmid pmirGLO-CC (*p* < 0.01) (Figure 4b), suggesting that the C>T mutation increased the luciferase activity of the *PLAG1* gene.

### 3.4. Role of miR-139 in Regulating the Post-Transcriptional Activity of the PLAG1 Gene

Notably, miRbase prediction indicated that the g.8795C>T mutation is located in the seed sequence of miR-139. To determine whether miR-139 affects *PLAG1* post-transcriptional activity, mimics and inhibitors of miR-139 were synthesized. Subsequently, pmirGLO-CC or pmirGLO-TT and miR-139 mimics or inhibitors were co-transfected into 293T cells. The results of the luciferase assay (Figure 4c) indicated that miR-139 mimics co-transfected with pmirGLO-CC or pmirGLO-TT significantly reduced luciferase activity, and the luciferase activity of pmirGLO-CC was lower than that of pmirGLO-TT. By contrast, miR-139 inhibitor co-transfected with pmirGLO-TT or pmirGLO-CC significantly increased luciferase activity (Figure 4d). These results revealed that the g.8795C>T mutation weakened the binding activity of miR-139, further increasing the luciferase activity of the *PLAG1* gene.

### 3.5. Correlation between miR-139 and PLAG1 Gene Expression

To confirm the correlation between miR-139 and *PLAG1* gene expression in the muscles of Hu sheep fetuses, real-time PCR was used to detect miR-139 and *PLAG1* expression levels. Correlation analysis showed that the expression of *PLAG1* (Figure 5) was downregulated with the upregulation of miR-139, suggesting that miR-139 was negatively correlated with the *PLAG1* gene.

## 4. Discussion

*PLAG1* plays an important role in animal growth, affecting height in humans [7,15,16,21], growth in mice, pigs, and cattle, and the shape of horses [10,11,18,22]. Moreover, *PLAG1* SNPs regulate growth traits in animals. Previous studies have shown that the g.48308 C>T mutation of *PLAG1* is associated with body height, chest circumference, and body length [23]; the T>G mutation at the rs109815800 locus increases the body height of Chinese local cattle [24]; and a 19 bp deletion of *PLAG1* has been shown to be significantly associated with growth traits in cattle breeds, with the hip width and rump length of Pinan cattle, heart girth and cannon bone circumference of Xianan cattle, as well as the heart girth, hip width, hucklebone width, rump length, height at the sacrum, and chest depth of Jiaxian cattle [25]. The g.48038 C >T polymorphism of *PLAG1* has been shown to be significantly associated with the body length of Bali cattle, and cattle with the CC genotype had a greater body length than the other two genotypes [26]. In goats, a 15-bp inDel mutation of *PLAG1* has been associated with the regulation of important growth characteristics of both adult and lamb goats, which may serve as an efficient molecular marker for goat breeding [27]. In sheep, Pan et al. [28] identified two indel variants (P2-del 30 bp and P4-del 45 bp) of the *PLAG1* gene that were significantly related to 15 growth traits of the Chinese Luxi blackhead sheep, suggesting that the two indel mutations were molecular markers for the selection of economic traits in sheep. In this study, we first obtained the complete 3′-UTR sequence and then identified the g.8795 C>T mutation in this region in a Hu sheep population. Correlation analysis revealed that the g.8795 C>T mutation was closely related to the early weight of Hu lambs, and the birth weight and weaning weight of the CT genotype were higher than those of the CC genotype for ram lambs, ewe lambs, twin populations, and the entire Hu lamb population, indicating that the g.8795C>T mutation is involved in regulating the early weight of Hu sheep.

The luciferase assay showed that the g.8795 C>T mutation may affect the post-transcriptional activity of the *PLAG1* gene, and miRNA prediction revealed that this mutation was located in the seed sequence of miR-139. It is well known that miRNAs play a role in post-transcriptional regulation in mammals [29,30]. For example, miR-10b-5p partially regulates the proliferation and differentiation of C2C12 myoblasts by directly targeting the 3′-untranslated region of NFAT5 [31]. Moreover, the rs1054564-C allele of GDF15 can stop has-miR-1233-3p-mediated translational suppression of GDF15 [32]. Exosomal miR-181a-5p activates the Wnt/β-catenin signaling pathway by targeting the Wnt inhibitor WIF1, thereby regulating proteins and genes related to hair follicle growth and development [33]. miR-142-5p targets FOXO3, promotes growth-related gene expression, and regulates skeletal muscle growth in chickens [34]. Furthermore, overexpression of miR-139 can suppress osteosarcoma cell growth, and loss of miR-139 can promote cell proliferation by regulating DNMT1 [35]. miR-139-5p, miR-940, and miR-193a-5p also inhibit the growth of hepatocellular carcinoma by targeting SPOCK1 [36].

The degree of base complementarity between miRNAs and target genes can affect the regulatory intensity of miRNAs and the activity of target genes [37,38]. Shi et al. [39] found that miR-181a, miR-135a, and miR-302c could not bind to the 3′-UTR sequence of the *PLAG1* gene after base mutation, and luciferase activity was significantly upregulated after mutation. A G>A mutation at the 3′-UTR of SLITRK1 increased the binding ability between miR-189 and SLITRK1, which led to the regulation of SLITRK1 by miR-189 [40]. In this study, we found that the g.8795C>T mutation weakened the binding between miR-139 and *PLAG1*, and miR-139 was involved in regulating the post-transcriptional activity of the *PLAG1* gene. Further analysis indicated that the expression level of miR-139 was negatively correlated with the expression of *PLAG1* in ovine muscles, suggesting that *PLAG1* may be the target gene of miR-139.

## 5. Conclusions

Our results revealed that the g.8795C>T locus located in the *PLAG1* 3’-UTR of Hu sheep can affect the birth weight and weaning weight of Hu sheep. We have discussed the possible mechanism from the perspective of the change in miRNA binding sequence caused by SNPs, suggesting that the g.8795C>T mutation weakens the binding capacity between miR-139 and the *PLAG1* gene, further increasing the post-transcriptional activity of PLAG1 and upregulating the early weight of Hu lambs (Figure 6). These results will be helpful in providing candidate molecular markers for the auxiliary selection of growth traits of Hu sheep, and accelerating the breeding progress of sheep breeds using Hu sheep as breeding material.

## Figures and Tables

**Figure 1 genes-14-00467-f001:**
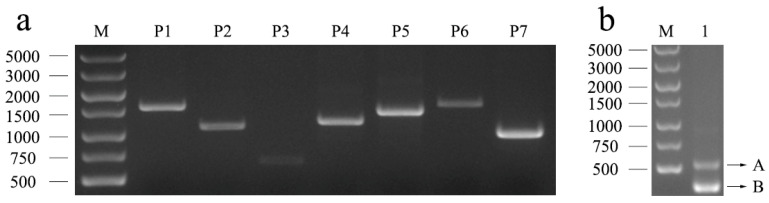
*PLAG1* 3′-UTR amplification in Hu sheep: Agarose electrophoresis of (**a**) primers P1, P2, P3, P4, P5, P6, and P7, respectively; (**b**) nested PCR (P9 primer) used to amplify the 3′-UTR of *PLAG1*. M: DNA marker (DL5000); 1: P9 amplification product.

**Figure 2 genes-14-00467-f002:**
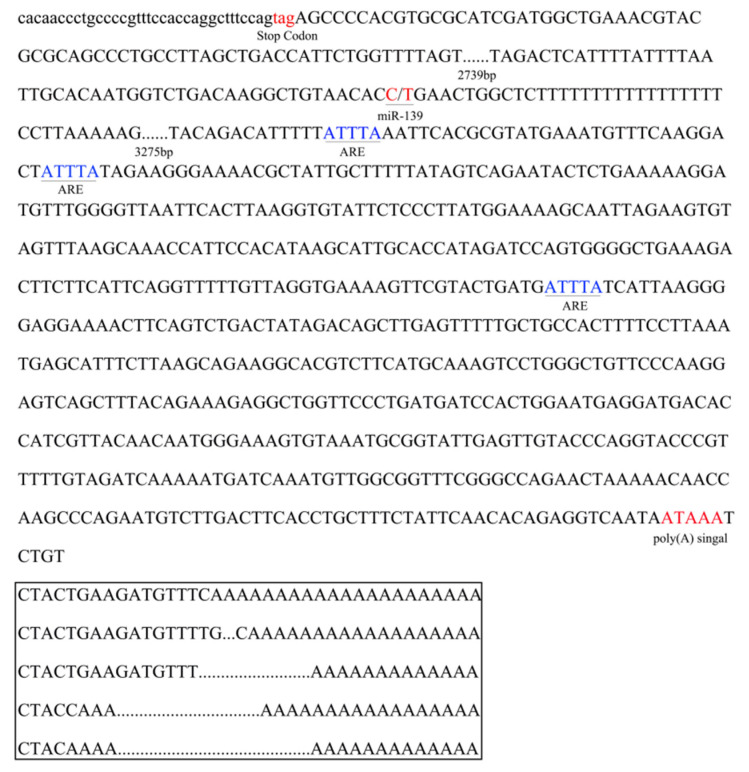
Characterization of the *PLAG1* 3′-UTR region in Hu sheep: black underlines represent AU-rich element (ARE) sequences; black boxes indicate the five different base sequences plus poly(A) tails; two parts (2739 bp and 3275 bp) of the sequence of *PLAG1* are omitted.

**Figure 3 genes-14-00467-f003:**
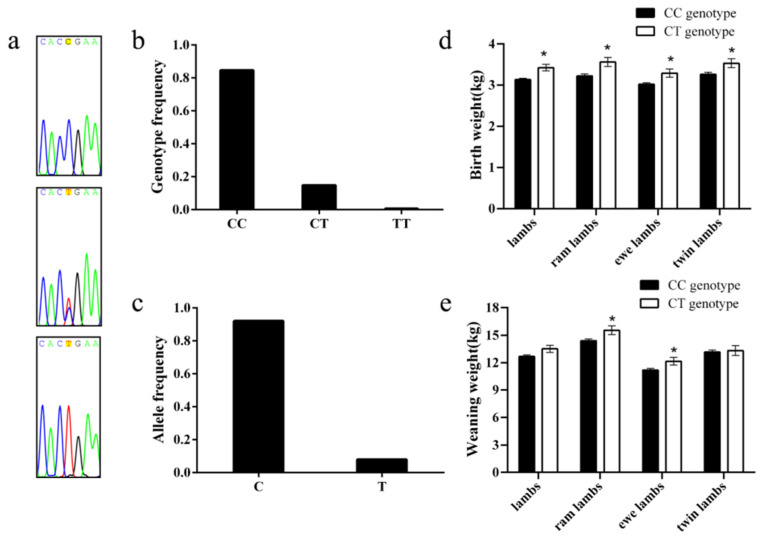
Mutation of g.8795 C>T in the 3′-untranslated region of the Hu sheep *PLAG1* gene and the relationship between different genotypes and early body weight: (**a**) sequence of different genotypes at the g.8795 C>T mutation; (**b**) genotype frequency; (**c**) allele frequency; (**d**) birth weight; (**e**) weaning weight. *, *p* < 0.05.

**Figure 4 genes-14-00467-f004:**
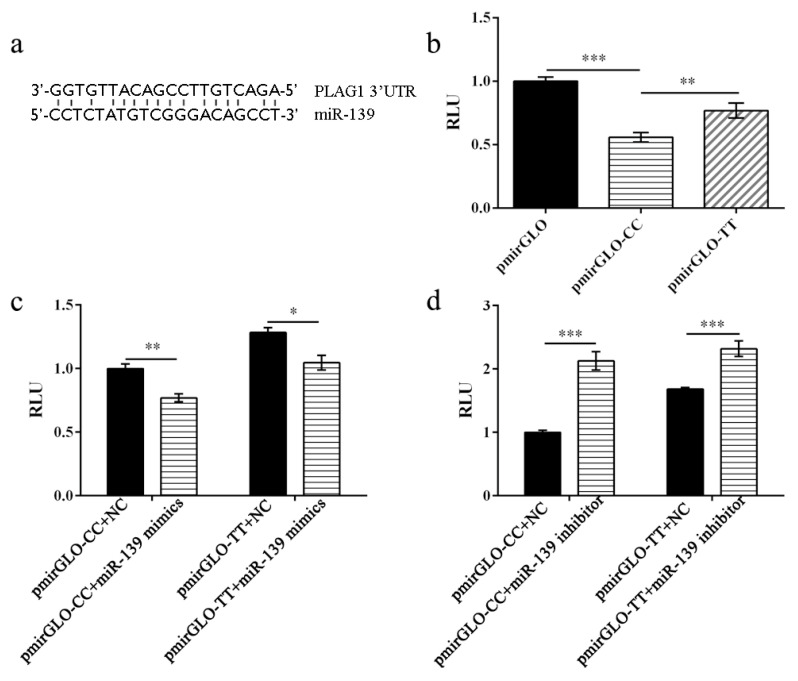
miR-139 downregulates the activity of the *PLAG1* gene: (**a**) predicted diagram of the miR-139 binding site sequence in *PLAG1* 3′-UTR; (**b**) dual-luciferase reporter analysis showed that g.8795C>T mutation affected the activity of *PLAG1*; (**c**) miR-139 mimics downregulated the luciferase activity of the *PLAG1* gene, and the luciferase activity decreased more when miR-139 mimics were co-transfected with allele CC than with allele TT; (**d**) miR-139 inhibitor upregulated the luciferase activity of the *PLAG1* gene. RLU: relative luciferase activity; *, *p* < 0.05; **, *p* < 0.01; ***, *p* < 0.001.

**Figure 5 genes-14-00467-f005:**
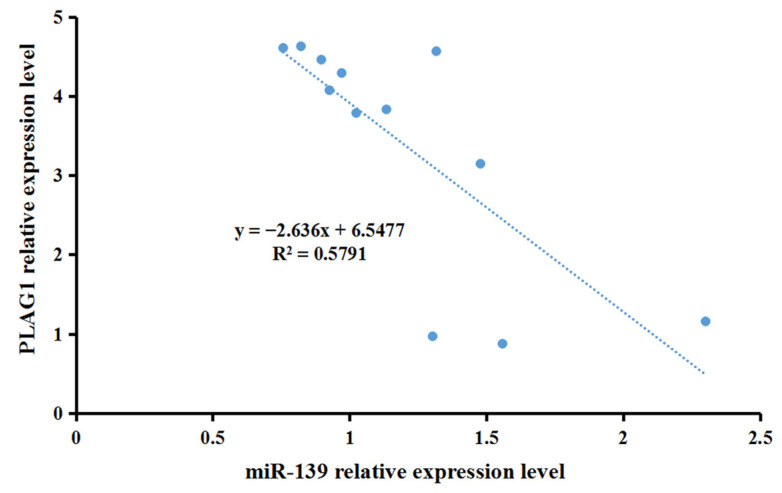
Negative correlation between miR-139 expression and *PLAG1* gene expression in fetal ovine muscle tissues.

**Figure 6 genes-14-00467-f006:**
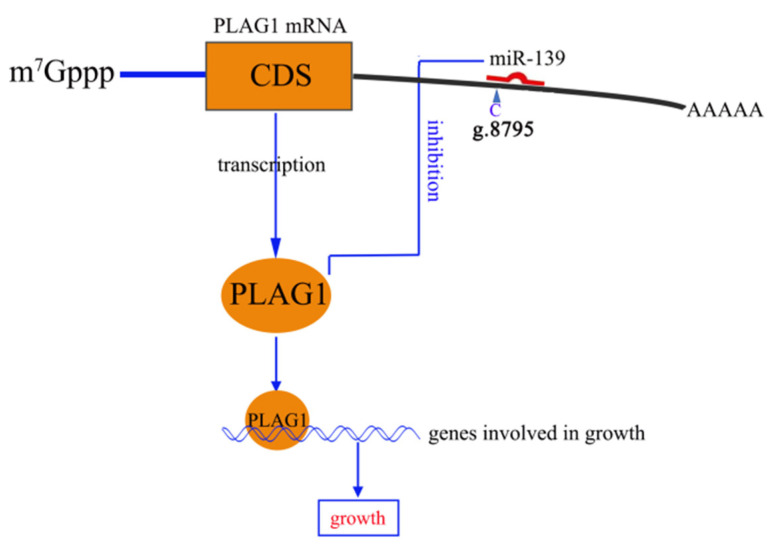
Schematic model depicting the proposed regulation pathway of miR-139-mediated PLAG1 transcription and further growth regulation of Hu sheep: the g.8795C>T mutation weakens the binding activity between miR-139 and PLAG1 3′-UTR, resulting in decreased inhibition of PLAG1 transcription, further increasing the birth and weaning weights of Hu sheep.

**Table 1 genes-14-00467-t001:** List of primers and oligonucleotides used in this study.

Number	Gene Name	Primer Sequence (5′–3′)	Annealing Temperature (°C)	Fragment Length (bp)	Useful
P1	*PLAG1*	F: ACCCGTTCAGTTCTACCTCAT R: CGTGGTTCCCAGACAAGTC	56	1529	PCR amplification and SNP identification
P2	*PLAG1*	F: AGCGCACCAGTATTTGTAGCA R: ACATGGAAATCCGCAGTGATA	56	1131	PCR amplification and SNP identification
P3	*PLAG1*	F: GTTTGAGGAGGGAGGGTTTAT R: CTCGACGGTGATTAAAGCAAT	56	658	PCR amplification and SNP identification
P4	*PLAG1*	F: CTGCCCGCTCTAGTTTCTAT R: GTCAGCTCTGGCTCATGTTT	56	1256	PCR amplification and SNP identification
P5	*PLAG1*	F: TTTGCCGACGTGTTGCTTGT R: CCGAATGGATGCCCAGTTTT	57	1473	PCR amplification and SNP identification
P6	*PLAG1*	F: TACAGATGACCCAGAATGAATG R: TGAAAGAGGTGCTATGAGAAAT	56	1599	PCR amplification and SNP identification
P7	*PLAG1*	F: TCCCTTGGCATTTACTGTCTG R: ACATTCTGGGCTTGGTTGTTT	54	1032	PCR amplification and SNP identification
P8	*PLAG1*	F: CAAACCATTCCACATAAGCATTGCACCAT R: CTAATACGACTCACTATAGGGCAAGCAGTGGTATCAACGCAGAGT(UPM)	71	519	3′ RACE
P9	*PLAG1*	F: GCAAAGTCCTGGGCTGTT R: CTAATACGACTCACTATAGGGC(shortP)	56	294	3′ RACE
P10	*PLAG1*	F: TGAAGAAGAGCCACAACCAG R: CTTGATGGGCACCGACAC	58	109	Real-time PCR
P11	β-Actin	F: CAGCCATCTTCTTGGGTAT R: CTGTGATCTCCTTCTGCATCC	60	150	Real-time PCR
P12	miR-139	F: GCCGAGTGGAGACGCGGCCCT R: CCAGCCACAAAAGAGCACAAT	60		Real-time PCR
P13	U6	F: CTCGCTTCGGCAGCACA R: AACGCTTCACGAATTTGCGT	60	94	Real-time PCR
P14	Mimics NC	F: UUCUCCGAACGUGUCACGUTT R: ACGUGACACGUUCGGAGAATT			Cell transfection
P15	miR-139 mimics	F: UGGAGAUACAGCCCUGUUGGAAU R: UCCAACAGGGCUGUAUCUCCAUU			Cell transfection
P16	Inhibitor NC	F: CAGUACUUUUGUGUAGUACAA			Cell transfection
P17	miR-139 inhibitor	F: AUUCCAACAGGGCUGUAUCUCCA			Cell transfection
P18	miR-139 (stem-loop)	CCTGTTGTCTCCAGCCACAAAAGAGCACAATATTTCAGGAGACAACAGGACTCCAAC			Reverse transcription

## Data Availability

The relevant data are contained in the article and the datasets used during the current study are available from the corresponding author on reasonable request.

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
