# Peer review of "PLAG1 g.8795C>T Mutation Regulates Early Body Weight in Hu Sheep by Weakening miR-139 Binding"

_genes, 2023, doi:10.3390/genes14020467_

Round 1
Reviewer 1 Report
Dear Author,
Reviewer #: MAJOR COMMENTS
The manuscript entitled "PLAG1 g.8795 C>T mutation regulates the early body weight by 2 miR-139 in Hu sheep" is an original and interesting work. As well, it has importance in its field of study. The methods presented are reliable and the results are also well explained. However, the main issue with this MS is correlated with language, so the authors are strongly asked to be supported by a native speaker to enhance the language quality of this work. Please refer to the following points for minor revisions of the manuscript.
SPECIFIC COMMENTS
Keywords:
Alphabetically arranged.
Introduction:
Page 1, Line 37-39, replace the sentences from, therefore, to traits of sheep with this sentence (Body weight is crucial selection criteria for improving livestock production. The weight of animals significantly affects farmer or breeder income (Ajafar et al., 2022; Al-Thuwaini and Al-Hadi, 2022).
Ajafar, M. H., Al-Thuwaini, T. M., & Dakhel, H. H. (2022). Association of OLR1 gene polymorphism with live body weight and body morphometric traits in Awassi ewes. Molecular Biology Reports, 1-5.
Al-Thuwaini, T. M., & Al-Hadi, A. B. A. (2022). Association of lamb sex with body measurements in single and twin on the Awassi ewes. Adv. Anim. Vet. Sci, 10(8), 1849-1853.
Materials and methods
2.1 Samples
- Need rearranged, added data collection with animals, and mentioned more details about birth weight and weaning weight, number of animals. How the animals were selected for the study. Did the animals where fed individually or the diet figures are averages?
- Primers 10 to 17 in Table 1 did not mention methods.
2.8 Statistical analyses
- Added more details and added the correlation analysis.
Discussion
Do not mention the figure in the discussion and moved this section from page 8, lines 24 to 25 (Therefore to Hu lambs) to the introduction.
Author Response
Dear Reviewer
Thank you very much for reviewing my submission. According to your suggestion and questions, we had revised this submission point by point.
Q1: The manuscript entitled "PLAG1 g.8795 C>T mutation regulates the early body weight by miR-139 in Hu sheep" is an original and interesting work. As well, it has importance in its field of study. The methods presented are reliable and the results are also well explained. However, the main issue with this MS is correlated with language, so the authors are strongly asked to be supported by a native speaker to enhance the language quality of this work. Please refer to the following points for minor revisions of the manuscript.
A1: Thanks for your suggestion, we had invited a professional editing service to make a comprehensive revision of this manuscript including the grammar, spelling and tense.
Q2: Keywords: Alphabetically arranged.
A2: Based on your suggestion, we've rearranged the keywords in alphabetical order.
Q3: Introduction:
Page 1, Line 37-39, replace the sentences from, therefore, to traits of sheep with this sentence (Body weight is crucial selection criteria for improving livestock production. The weight of animals significantly affects farmer or breeder income (Ajafar et al., 2022; Al-Thuwaini and Al-Hadi, 2022).
Ajafar, M. H., Al-Thuwaini, T. M., & Dakhel, H. H. (2022). Association of OLR1 gene polymorphism with live body weight and body morphometric traits in Awassi ewes. Molecular Biology Reports, 1-5.
Al-Thuwaini, T. M., & Al-Hadi, A. B. A. (2022). Association of lamb sex with body measurements in single and twin on the Awassi ewes. Adv. Anim. Vet. Sci, 10(8), 1849-1853.
A3: Thanks for your suggestion, we've replaced the sentences from, therefore, to traits of sheep with this sentence (Body weight is crucial selection criteria for improving livestock production. The weight of animals significantly affects farmer or breeder income) and added the two references (Ajafar et al., 2022; Al-Thuwaini and Al-Hadi, 2022) in corresponding place.
Q4: Materials and methods
2.1 Samples
Need rearranged, added data collection with animals, and mentioned more details about birth weight and weaning weight, number of animals. How the animals were selected for the study.
A4: Based on your suggestion, we added these data (The 352 lambs with the same herd, same environment and diet, the similar ages at the same season were selected from the Xilaiyuan Sheep Industry Co., Ltd., along with corresponding records of birth weight and weaning weight (45days after birth) ) in corresponding place.
Q5: Did the animals where fed individually or the diet figures are averages?
A5: These sheep were fed freely.
Q6: Primers 10 to 17 in Table 1 did not mention methods.
A6: Based on your suggestion, we explained how to use P10 - P17 primers in “real-time PCR and plasmid construction and synthesis” part.
Q7: 2.8 Statistical analyses
Added more details and added the correlation analysis.
A7: Based on your suggestion, we added the analysis model in “Statistical analyses” part. The contents were as follows: The correlation analysis model was identified by General Linear Model (GLM) procedure using SPSS software (version 16.0; SPSS Inc, USA). The influencing factors in this study included the birth status (simple or twin), maternal ages (two levels, 2-3), and sex (male and female) and genotype (CC and CT); Yijkl = μ + Gi + Bj + Mk + Sl + eijkl. Where, Yijkl=the trait measured on each of the ijk animal, μ=the overall population mean, Gi=the fixed effect associated with genotype, Bj=the fixed effect of birth status, Mk= the fixed effect of maternal ages, Sl=the fixed effect of sex, eijkl=random residual error.
Q8: Discussion
Do not mention the figure in the discussion and moved this section from page 8, lines 24 to 25 (Therefore to Hu lambs) to the introduction.
A8: Based on your suggestion, we didn't mention figure in our discussion and move this section from page 8, lines 24 to 25 (Therefore to Hu lambs) to page 2, lines 73 to 76 (Combining to Hu lambs).
All changes were marked in red font.
If have any question, please contact with me.
Best wishes.
Yours,
Yue Wang
Reviewer 2 Report
1. Hypothesis of present study is not clear. Make it more clear in Introduction part.
2. Association of targeted SNP with early growth traits need to be performed using suitable model taking into consideration various parameters.
3. How many samples processed for sequencing not mentioned.
4. Gene and genotypic freq. of targeted loci need to performed.
5. Authors tried to study the effect of targeted mutation on post transcription factor binding sites. However, based on single SNP identification of mirna binding site is not reliable.
6. The author needs to perform NGS or microarray panel analysis for more informative and validated results.
7. Author need to redesign experiment for better results and interpretation.
Author Response
Dear Reviewer
Thank you very much for reviewing my submission. According to your suggestion and questions, we had revised this submission point by point.
Q1: Hypothesis of present study is not clear. Make it more clear in Introduction part.
A1: Thanks for your suggestion. In order to present the research hypothesis more clearly, we had put forward more specific assumptions and corresponding results in the last paragraph of the introduction.
Q2: Association of targeted SNP with early growth traits need to be performed using suitable model taking into consideration various parameters.
A2: Yes, you are correct. We supplemented the analysis model in the “Statistical analyses” section, including environmental factors, mother’s age and birth parity, genotype, number of siblings and other factors.
Q3: How many samples processed for sequencing not mentioned.
A3: Thanks for your suggestion. We mentioned that the number of samples sequenced was 352 in line 83.
Q4: Gene and genotypic freq. of targeted loci need to performed.
A4: Thanks for your suggestion. The frequencies of allele C and T, the frequencies of genotype CC, CT, TT were showed in fig 3b and fig 3c.
Q5: Authors tried to study the effect of targeted mutation on post transcription factor binding sites. However, based on single SNP identification of mirna binding site is not reliable.
A5: Thanks very much, it is a good question. Based on our results and literature reciew, the degree of base complementary between miRNAs and target genes can affect the regulatory intensity of miRNA and further affect the activity of target genes (Doench et al., 2004; Lewis et al., 2005). Abelson et al. Found that a SNP of A>G mutation in SLITRK1 3’UTR strengthened the effect of miR-189 on regulating SLITRK1(Abelson et al., 2005). A G to A transition in the 3’UTR of GDF8 creates a target site for miR1 and miR206, which causing translational inhibition of GDF8 gene and hence contributes to the muscle hypertrophy of Texel sheep (Clop et al., 2006); In our study, the g.8795 C>T mutation happened in the seed sequences of miR-139, after a base mutation, the binding ability weakened between miR-139 and PLAG1, and miR-139 was involved in regulating the post-transcriptional activity of PLAG1 gene.
- Abelson, J.F.; Kwan, K.Y.; O'RoakB, J.; BaekD, Y.; Stillman, A.A.; Morgan, T.M.; Matjews, C.A.; Pauls, D.L.; Rasin, M.R.; Gunel, M.; Davis, N.R.; Ercan-Sencicek, A.G.; Guez, D.H.; Spertus, J.A.; Leckman, J.F.; Dure 4th, L.S.; Kurlan, R.; Singer, H.S.; Gilbert, D.L.; Farhi, A.; Louvi, A.; Lifton, R.P.; Sestan, N.; State, M.W. Sequence variants in SLITRK1 are associated with Tourette's syndrome. Science 2005, 310, 317-320.
- Clop A, Marcq F, Takeda H, et al. A mutation creating a potential illegitimate microRNA target site in the myostatin gene affects muscularity in sheep[J]. Nat Genet, 2006, 38:813-818
- Shi, L.; Li, X.; Wu, Z.Q.; Li, X.L.; Nie, J.; Guo, M.Z.; Mei, Q.; Han, W. DNA methylation-mediated repression of miR-181a/135a/302c expression promotes the microsatellite-unstable colorectal cancer development and 5-FU resistance via targeting PLAG1. J. Genet. Genomics. 2018, 45, 205- 214.
- Sun Y, Lu F, Han X, et al. MiR-424 and miR-27a increase TRAIL sensitivity of acute myeloid leukemia by targeting PLAG1[J]. Oncotarget, 2016, 7:25276-25290.
- Doench, J.G.; Sharp, P.A. Specificity of microRNA target selection in translational repression. Gene. Dev. 2004, 18, 504-511.
- Lewis, B.P.; Burge, C.B.; Bartel, D.P. Conserved seed pairing, often flanked by adenosines, indicates that thousands of human genes are microRNA targets. Cell 2005, 120, 15-20.
Q6: The author needs to perform NGS or microarray panel analysis for more informative and validated results.
A6: Thanks for your good suggestion. PLAG1 plays an important role in regulating body weight and body length in domestic animals by SNP array and genome-wide association. In cattle, An et al.(2019) performed a genome-wide association study and found PLAG1 gene was associated with body measurement traits in 463 Wagyu beef cattle typed with the IIlumina Bovine HD 770k SNP array. In pigs, PLAG1 was identified as one of the candidate gene associated with limb bone length using SNP60 BeadChip in 2004 pigs (Guo et al., 2015). Qiao et al (2015) performed a GWAS to analyze 22 traits related to growth and fatness on White Duroc × Erhualian F2 population and a Chinese Sutai half-sib population, and found PLAG1 was proximal to the top SNP and stand out as a strong candidate gene.
So in this study, we intended to verify PLAG1gene function on growth traits in Hu sheep. And the results showed that PLAG1’s function on sheep were consistent with those of pigs and cattle, indicating that PLAG1 was a candidate molecular marker for regulating growth of domestic animals.
1. An B, Xia J, Chang T, et al. Genome-wide association study reveals candidate genes associated with body measurement traits in Chinese Wagyu beef cattle. Anim Genet, 2019, 50(4): 386-390.
2. Guo Y, Hou L, Zhang X, et al. A meta analysis of genome-wide association studies for limb bone length in four pig populations. BMC Genet, 2015, 16: 95.
3. Qiao R, Gao J, Zhang Z, et al. Genome-wide association analyses reveal significant loci and strong candidate genes for growth and fatness traits in two pig populations. Genet Sel Evol, 2015, 47(1): 17.
Q7: Author need to redesign experiment for better results and interpretation.
A7: Thanks for your suggestion. In this study, we firstly found that PLAG1 was a candidate gene for domestic animals growth through literature review, but there was no relevant research on sheep. Further research found the sequence of PLAG1 on sheep was still incomplete. Based on these, we amplified its sequence and then screened polymorphisms in Hu sheep population and analyzed its association with growth traits. Moreover, g.8795 C>T mutation was identified and significantly related to early body weight. At last, we explored this regulation mechanism using software prediction combined with experimental verification.
All changes were marked in red font.
If have any question, please contact with me.
Best wishes.
Yours,
Yue Wang

Reviewer 3 Report
This is an interesting and well-written manuscript aimed to identify the complete 3’-untranslated region (3’ UTR) of the ovine PLAG1 gene and screens for the relationships between polymorphisms and early body weight using DNA sequencing. Results are clear and concise, and discussion is also clear and well supported. I only recommend to provide more information about some molecular methods and to consider next minor corrections:
- Line 55: The reference number of Littlejohn et al. is missing.
- Line 68: Replace “identifies” by “identified”.
- Line 71: Replace “are” by “were”.
- Line 79: Separate round bracket from the text.
- Line 99: Replace “primers showed” by “primers is showed”.
- Line 107: Please describe briefly the genotyping process.
- Line 112: Please describe briefly the rtPCR protocol.
- Line 115: Please describe briefly the process to predict miRNA binding sites.
- Line 129: Replace “are” by “were”.
- Line 132: Please indicate the p-value threshold for the statistical tests.
- Line 243: Replace “was” by “had a”.
- Lines 245-247: I suggest writing the sentence in past tense.
- Lines 254-259: The sentence is too long. I suggest separating it in two shorter sentences.
- Line 278: Remove the period sign after “Genomics”.
- Line 355: Remove the period sign after “Genomics”.
Author Response
Dear Reviewer
Thank you very much for reviewing my submission. According to your suggestion and questions, we had revised this submission point by point.
Q1:
Line 55: The reference number of Littlejohn et al. is missing.
Line 68: Replace “identifies” by “identified”.
Line 71: Replace “are” by “were”.
Line 79: Separate round bracket from the text.
Line 99: Replace “primers showed ” by “primers is showed ”.
A1: Based on your suggestion, we are done with the changes.
Q2: Line 107: Please describe briefly the genotyping process.
A2: Based on your suggestion, we have described the genotyping process briefly from line 122 to 123.
Q3: Line 112: Please describe briefly the rtPCR protocol.
A3: Based on your suggestion, we have described the rtPCR protocol briefly from line 127 to 140.
Q4: Line 115: Please describe briefly the process to predict miRNA binding sites.
A4: Based on your suggestion, we have described the process of predicting miRNA binding sites from line 142 to 146.
Q5:
Line 129: Replace “are” by “were”.
Line 243: Replace “was” by “had a”.
Lines 245-247: I suggest writing the sentence in past tense.
Lines 254-259: The sentence is too long. I suggest separating it in two shorter sentences.
Line 278: Remove the period sign after “Genomics”.
Line 355: Remove the period sign after “Genomics”.
A5: Based on your suggestion, we are done with the changes.
Q6: Line 132: Please indicate the p-value threshold for the statistical tests.
A6: Based on your suggestion, we have described the p-value threshold in the part of statistical analyses.
All changes were marked in red font.
If have any question, please contact with me.
Best wishes.
Yours,
Yue Wang
